# Hyperuricemia Is Associated with Left Ventricular Dysfunction and Inappropriate Left Ventricular Mass in Chronic Kidney Disease

**DOI:** 10.3390/diagnostics10080514

**Published:** 2020-07-24

**Authors:** Tai-Hua Chiu, Pei-Yu Wu, Jiun-Chi Huang, Ho-Ming Su, Szu-Chia Chen, Jer-Ming Chang, Hung-Chun Chen

**Affiliations:** 1Department of General Medicine, Kaohsiung Medical University Hospital, Kaohsiung 807, Taiwan; tata14080222@gmail.com; 2Division of Nephrology, Department of Internal Medicine, Kaohsiung Medical University Hospital, Kaohsiung Medical University, Kaohsiung 807, Taiwan; wpuw17@gmail.com (P.-Y.W.); karajan77@gmail.com (J.-C.H.); jemich@kmu.edu.tw (J.-M.C.); 3Department of Internal Medicine, Kaohsiung Municipal Siaogang Hospital, Kaohsiung Medical University, Kaohsiung 812, Taiwan; cobeshm@seed.net.tw; 4Faculty of Medicine, College of Medicine, Kaohsiung Medical University, Kaohsiung 807, Taiwan; 5Division of Cardiology, Department of Internal Medicine, Kaohsiung Medical University Hospital, Kaohsiung Medical University, Kaohsiung 807, Taiwan; 6Research Center for Environmental Medicine, Kaohsiung Medical University, Kaohsiung 807, Taiwan

**Keywords:** uric acid, left ventricular ejection fraction, left atrial diameter, inappropriate left ventricular mass, chronic kidney disease

## Abstract

Background. Hyperuricemia is common in patients with chronic kidney disease (CKD), and this may lead to poor cardiovascular (CV) outcomes. The aim of this cross-sectional study was to assess associations among serum uric acid (UA) and echocardiographic parameters, ankle-brachial index (ABI), and brachial-ankle pulse wave velocity (baPWV) in patients with CKD. Methods. A total of 418 patients with CKD were included. The echocardiographic measurements included left atrial diameter (LAD), left ventricular ejection fraction (LVEF) and the ratio of observed to predict left ventricular mass (LVM). ABI, baPWV and medical records were obtained. Results. Multivariable forward logistic regression analysis showed that a high UA level was significantly associated with LAD > 47 mm (odds ratio [OR], 1.329; *p* = 0.002), observed/predicted LVM > 128% (OR, 1.198; *p* = 0.008) and LVEF < 50% (OR, 1.316; *p* = 0.002). No significant associations were found between UA and ABI < 0.9 or baPWV > 1822 cm/s. Multivariate stepwise linear regression analysis showed that a high UA level correlated with high LAD (unstandardized coefficient β, 0.767; *p* < 0.001), high observed/predicted LVM (unstandardized coefficient β, 4.791; *p* < 0.001) and low LVEF (unstandardized coefficient β, −1.126; *p* = 0.001). No significant associations between UA and low ABI and high baPWV were found. Conclusion. A high serum UA level was associated with a high LAD, high observed/predicted LVM and low LVEF in the patients with CKD. A high serum UA level may be correlated with abnormal echocardiographic parameters in patients with stage 3–5 CKD.

## 1. Introduction

Cardiovascular (CV) abnormalities such as left ventricular hypertrophy (LVH) and left ventricular (LV) dysfunction are common in patients with chronic kidney disease (CKD) due to chronic inflammation, activation of the renin-angiotensin-aldosterone system and changes in hemodynamics [1]. In addition, patients with CKD have high rates of atherosclerosis, and the progression of atherosclerosis in these patients has been closely associated with a decline in kidney function [2]. Interactions between the CV system and kidneys are bidirectional [1,3]. Thus, it is very important to recognize these problems and provide timely interventions to break this vicious circle.

Left atrial size, which reflects LV diastolic dysfunction, has been associated with various adverse CV events [4], and left atrial diameter (LAD) has been shown to be an easily accessible marker to measure the size of the left atrium [5]. The condition of increased left ventricular mass (LVM) exceeding the compensatory hemodynamic need is known as inappropriate LVM (iLVM) [6]. The ratio of observed to predicted left ventricular mass (o/p LVM) is used to assess the ‘appropriateness’ of LVM, where the predicted LVM is based on sex, body size, and cardiac workload. The iLVM was reported to be a better index in measuring myocardial function than the traditional LVM index [6]. 

Uric acid (UA) is the end product of purine metabolism in humans, and approximately 70% of UA load is eliminated through the kidneys [7]. In the general population, hyperuricemia has been associated with many CV anomalies, including increased local arterial stiffness, carotid intima-media thickness, LVH, and atherosclerosis [8,9,10,11,12]. In patients with CKD, hyperuricemia has also been associated with many adverse CV outcomes, and high rates of CV events and CV mortality [13,14,15]. Taiwan has the highest incidence and prevalence rates of end-stage renal disease worldwide [16], however data on the association between the level of serum UA and CV risk factors, including echocardiographic abnormalities and subclinical atherosclerosis, in patients with CKD are limited. Therefore, the aim of this study was to investigate the associations among serum UA level and echocardiographic parameters, ankle-brachial index (ABI), and brachial-ankle pulse wave velocity (baPWV), both of which are indicators of atherosclerosis and arterial stiffness, in patients with CKD.

## 2. Subjects and Methods

### 2.1. Study Patients and Design

The study was conducted at a regional hospital in southern Taiwan, and consecutively enrolled 540 pre-dialysis patients with CKD stages 3 to 5 according to the National Kidney Foundation-Kidney Disease Outcomes Quality Initiative (K/DOQI) guidelines from our Outpatient Department of Internal Medicine from January 2007 to May 2010 [17]. We excluded patients with significant atrial fibrillation, mitral and aortic valvular diseases, and those with inadequate imaging findings. Consequently, a total of 418 patients (mean age 65.8 ± 12.7 years, 230 males/188 females) were included. The study protocol was approved by the Institutional Review Board of our hospital, (approved code KMUH-IRB-20110158, approved on 23 June 2011), and all of the included patients provided written informed consent.

### 2.2. Evaluation of Cardiac Structure and Function

One experienced cardiologist blind to all other data performed all of the echocardiographic examinations using a VIVID 7 system (General Electric Medical Systems, Horten, Norway). During the examinations, the patients were placed in the left decubitus position and asked to breathe quietly. Two-dimensional M-mode images were recorded from standardized views. The echocardiographic measurements included LV posterior wall thickness in diastole (LVPWTd), LV internal diameter in diastole (LVIDd), interventricular septal wall thickness in diastole (IVSTd), and left atrial diameter (LAD). LV systolic function was assessed by LV ejection fraction (LVEF). Observed LV mass (LVM) was calculated using the modified Devereux method, i.e., LVM = 1.04 × [(IVSTd + LVIDd + LVPWTd)^3^ − LVIDd^3^] − 13.6 g [18]. Predicted LVM was estimated using the following equation [19]: predicted LVM = 55.37 + 6.64 × height (m^2.7^) + 0.64 × stroke work − 18.07 × sex (with male = 1 and female = 2). Stroke work (in gram meters) was estimated as systolic blood pressure × stroke volume product × 0.0144. Inappropriate LVM was defined as when observed/predicted LVM > 128%, i.e., when the observed LVM was >28% of the predicted value [19,20].

### 2.3. Assessment of baPWV and ABI

baPWV and ABI were measured using an ABI-form device that allowed for simultaneous blood pressure measurements in the ankles and arms. Occlusion and monitoring cuffs were placed snugly around both upper arms and lower extremities with the patients placed in the supine position. ABI was calculated as ankle/arm systolic blood pressure using the lowest ankle systolic blood pressure value. baPWV was calculated as transmission distance/transmission time. The lowest of the bilateral ABI values and highest of baPWV values were used for each patient. Peripheral artery occlusive disease was defined as an ABI < 0.9 in either leg.

### 2.4. Collection of Demographic, Medical, and Laboratory Data

Demographic and medical data including age, sex and comorbidities were obtained from the patients’ medical records or interviews. Fasting blood samples were collected from all patients within 30 days of enrollment, and an autoanalyzer (Roche Diagnostics GmbH, D-68298 Mannheim COBAS Integra 400) was used to obtain laboratory data. This autoanalyzer was also used to measure serum creatinine level with the compensated Jaffé method using an isotope-dilution mass spectrometry-traceable calibrator [21]. Estimated glomerular filtration rate (eGFR) was calculated using the Modification of Diet in Renal Disease equation [22]. In addition, data on the antihypertensive medications taken during the study period were obtained and recorded from the patients’ medical records, and included β-blockers, angiotensin II receptor blockers (ARBs), angiotensin converting enzyme inhibitors (ACEIs), calcium channel blockers, diuretics and hypouricemic agents (allopurinol and benzbromarone).

### 2.5. Reproducibility

Thirty patients were randomly selected for evaluation of the reproducibility of LVM. Mean percent error was calculated as the absolute difference divided by the average of the two observations.

### 2.6. Statistical Analysis

Statistical analysis was performed using SPSS version 19.0 for Windows (SPSS Inc., Chicago, IL, USA). Data are expressed as percentage, mean ± standard deviation or median (25–75th percentile) for triglycerides. Differences between groups were analyzed using the chi-square test for categorical variables and the independent t-test for continuous variables. Multiple logistic and linear regression analyses were used to identify the factors associated with ABI, baPWV and echocardiographic parameters. A difference was considered significant if the *p* value was less than 0.05. 

## 3. Results

A total of 418 patients with CKD who were not receiving dialysis were included (mean age 65.8 ± 12.7 years; 230 males and 188 females). Sex-specific median UA values were used to stratify the patients into two groups (males: 7.3; females: 6.7 mg/dL). Comparisons of the clinical characteristics between the two study groups are shown in Table 1. Compared to the patients with a lower UA level, those with a higher UA level had a higher prevalence of gout and congestive heart failure, higher triglyceride levels, lower eGFR, used more ACEIs and/or ARBs, β-blockers and diuretics, or had higher LAD and higher observed/predicted LVM. Figure 1 illustrates the prevalence of LAD > 47 mm, observed/predicted LVM > 128%, LVEF < 50%, ABI < 0.9, and baPWV > 1822 cm/s (median) between the two study groups. The patients with a higher UA level had higher rates of LAD > 47 mm (5.1% vs. 11.4%, *p* = 0.017) and observed/predicted LVM > 128% (64.1% vs. 74.1%, *p* = 0.026), but not LVEF < 50% (13.8% vs. 18.9%, *p* = 0.160), ABI < 0.9 (8.3% vs. 10.4%, *p* = 0.450) or baPWV > 1822 cm/s (49.3% vs. 50.7%, *p* = 0.769).

### 3.1. Determinants of LAD > 47 mm, Observed/Predicted LVM > 128%, LVEF < 50%, ABI < 0.9, and baPWV > 1822 cm/s in the Study Patients

Table 2 shows the risk factors for LAD > 47 mm, observed/predicted LVM > 128%, LVEF < 50%, ABI < 0.9, and baPWV > 1822 cm/s. After adjusting for age, sex, diabetes mellitus, coronary artery disease, congestive heart failure, mean artery pressure, UA, fasting glucose, triglycerides, total cholesterol, hematocrit, eGFR and medications, multivariate forward logistic regression analysis showed that a high UA level (odds ratio [OR], 1.329; 95% confidence interval [CI], 1.111 to 1.590; *p* = 0.002) was significantly associated with LAD > 47 mm. In addition, diabetes mellitus, low mean artery pressure, high UA level (OR, 1.198; 95% CI, 1.048 to 1.369; *p* = 0.008), and the use of diuretics were significantly associated with observed/predicted LVM > 128%. Coronary artery disease, congestive heart failure, high UA level (OR, 1.231; 95% CI, 1.025 to 1.480; *p* = 0.026), high fasting glucose, and lower rates of the use of calcium channel blockers and diuretics were significantly correlated with LVEF < 50%. Old age, high fasting glucose, and the use of ACEIs and/or ARBs were significantly associated with ABI < 0.9. Finally, old age, diabetes mellitus, congestive heart failure, high mean artery pressure, and high fasting glucose were significantly associated with baPWV > 1822 cm/s.

### 3.2. Correlation between UA and Echocardiographic Parameters

Figure 2 illustrates the scatter plots of distribution and correlation of serum UA and LAD (A), observed/predicted LVM (B) and LVEF (C). The UA was positively correlated with LVEF (*r* = 0.228, *p* < 0.001) and observed/predicted LVM (*r* = 0.167, *p* = 0.001), and negatively correlated with LVEF (*r* = −0.127, *p* = 0.009).

### 3.3. Determinants of LAD, Observed/Predicted LVM, LVEF, ABI, and baPWV in the Study Patients

Table 3 summarizes the possible determinants of LAD, observed/predicted LVM, LVEF, ABI, and baPWV in our study patients. After adjusting for age, sex, diabetes mellitus, coronary artery disease, congestive heart failure, mean artery pressure, UA, fasting glucose, triglycerides, total cholesterol, hematocrit, eGFR and medications, multivariate stepwise linear regression analysis showed that congestive heart failure, a high UA level (unstandardized coefficient β, 0.767; 95% CI, 0.421 to 1.113; *p* < 0.001) and low eGFR were significantly correlated with high LAD. We have further analyzed the determinants of the ratio of LAD to body surface area (BSA) using multivariable stepwise linear regression analysis, and still found that high UA level (unstandardized coefficient β, 0.291; 95% CI, 0.065 to 0.517; *p* = 0.012) was significantly correlated with high LAD/BSA. 

Furthermore, young age, congestive heart failure, low mean artery pressure, high UA level (unstandardized coefficient β, 4.791; 95% CI, 2.471 to 7.111; *p* < 0.001) and the use of diuretics were significantly associated with high observed/predicted LVM. Coronary artery disease, congestive heart failure, high UA level (unstandardized coefficient β, −1.126; 95% CI, −1.763 to −0.489; *p* = 0.001), high fasting glucose, and lower rates of the use of calcium channel blockers and diuretics were significantly associated with a low LVEF. Old age, coronary artery disease, and congestive heart failure were significantly associated with a low ABI. Lastly, old age, diabetes mellitus, congestive heart failure, high mean artery pressure, and high fasting glucose were significantly associated with a high baPWV.

## 4. Discussion

In this study, we found significant associations among hyperuricemia and high LAD, high observed/predicted LVM and low LVEF. However, ABI and baPWV failed to show significant independent association with serum UA.

Enlargement of the left atrium has been associated with various CV diseases including atrial fibrillation [23], stroke [24], and heart failure [25]. In the present study, high UA was associated with a large LAD in the patients with CKD. This finding is in agreement with a study by Gromadziński et al., who demonstrated that hyperuricemia was a risk factor for LV diastolic dysfunction in patients with CKD [26]. The possible mechanisms relating LA size to uric acid are as below. Left atrial (LA) remodeling and enlargement may result from pressure or volume overload in the left atrium, reflecting LV diastolic dysfunction or mitral valve disease [4]. Previous studies have demonstrated association between high UA and a large LAD and LV diastolic dysfunction in patients with CKD [26], newly diagnosed heart failure [27], atrial fibrillation [28], and dilated cardiomyopathy [29]. This may imply that LV diastolic dysfunction and resultant pressure or volume overload in the left atrium may be the link between uric acid and enlargement of LA. Besides, oxidative stress and inflammatory process in patients with hyperuricemia may also contribute to LA remodeling and enlargement [30,31]. At a neurohormonal level, another possible mechanism relating LA size to uric acid is that LA remodeling is the consequence of interstitial fibrosis, and is associated with stimulation of pro-fibrotic factors, such as angiotensin II and transforming growth factor beta [32]. 

Increasing clinical evidence has shown that serum UA plays a role in the development of LVH. In a study exploring the value of serum UA in predicting new-onset echocardiographic LVH over a 10-year follow-up period, Cuspidi et al. found that serum UA was a predictor of long-term echocardiographic changes from normal LVM index to LVH [33]. The link between serum UA and LVH may be due to its impact on hypertension, as UA has been associated with the risk of incident hypertension [34]. Some experimental in vivo studies have reported that UA may contribute to LVH through mechanisms such as increasing oxidative stress and endothelin-1 levels in the myocardium [35]. The iLVM is used to describe the condition of increased LVM exceeding the compensatory hemodynamic need [6]. Many researchers have reported that the iLVM can be used as an indicator of future CV risk or poor CV outcomes in different patient groups, including those with hypertension [36], CKD [37], and diabetes mellitus [38]. However, few studies have investigated the relationship between UA and iLVM. In the current study, we found that hyperuricemia was associated with a high o/p LVM in the patients with CKD. Previous studies have reported prevalence rates of iLVM ranging from 9–37% in different patient groups [36,39,40,41]. The prevalence of iLVM in our patients was as high as 64.9%. The high prevalence of iLVM in patients with CKD and the association between iLVM and impaired renal function have been reported [42]. 

In the present study, compared to the patients with a lower UA level, the group with a higher UA level had a higher prevalence of lower eGFR, used more ACEIs and/or ARBs, β-blockers and diuretics. Lower eGFR may contribute to lower UA clearance and higher serum UA level, and the use of antihypertensive medications may have influenced LV geometry and functional parameters. These factors may interfere in the impact of UA on echocardiographic parameters. To investigate the influence of the use of medications and eGFR, we included various classes of antihypertensive drugs and level of eGFR in the multivariate analysis, and still found that UA was significantly associated with abnormal echocardiographic parameters.

In the present study, we found an association between high serum UA level and low LVEF. Several epidemiological studies have also reported an association between UA and reduced LVEF. In a post hoc analysis of longitudinal data from the Framingham Offspring Study, Krishnan et al. reported that patients with serum UA > 6.2 mg/dL had an OR of 9.013 for abnormal LVEF [43]. In addition, Borghi et al. found that a high serum UA level was associated with a low LVEF in elderly male patients with heart failure [44]. The exact molecular mechanism underlying the damage-causing impact of UA on cardiac systolic function is still not well understood, although several possible theories have been proposed. The elevation of serum UA may indicate an underlying increase in the activity of xanthine oxidase, leading to the increased production of oxygen free radicals [45]. As a consequence, oxidative stress may be involved in the development of cardiac remodeling and dysfunction. In the present study, we also found that a high serum UA level was associated with a low LVEF in our patients with CKD. It is worth noting that in patients with CKD, hyperuricemia may cause unfavorable effects on the heart both through increased oxidative pressure, and also by damaging the kidneys and worsening the vicious circle between the heart and kidneys.

Atherosclerosis and vascular stiffness are common mechanisms in the pathogenesis of CV diseases, and can be assessed using the ABI and baPWV. In pre-dialysis patients with CKD stages 3–5, Yoshitomi et al. reported that ABI can predict CV events, mortality and all-cause mortality [46]. Atherosclerosis and vascular stiffness may play roles in the link between serum UA and CV abnormalities. However, in the present study, we did not find a significant association between serum UA and ABI or baPWV. The mechanism by which UA is linked to atherosclerosis is still under debate. In addition to oxidative stress and free radicals, a high UA level may impair the generation and release of nitric oxide and cause endothelial dysfunction [47]. Furthermore, a high serum UA level may activate hormonal vasoconstriction systems such as the renin-angiotensin system in vascular endothelial cells [48]. In a large cross-sectional study, Shankar et al. reported that higher serum UA levels were associated with peripheral arterial disease (defined as ABI < 0.9) in the US general population [49]. In addition, Song et al. reported that serum UA levels were associated with asymptomatic polyvascular stenosis, including peripheral arterial disease, in a prospective cohort study of 2644 adults [50]. Possible reasons for the difference between the results of the present study and previous studies are as follows. First, as some studies have reported, factors such as arterial calcification and concomitant clinical peripheral neuropathy may limit the diagnostic value of ABI in patients with diabetes mellitus [51]. The prevalence rates of diabetes mellitus in the present study were 30.9% and 38.3% in the lower and higher UA groups respectively. The relatively high prevalence of diabetes mellitus may have prevented the ABI from being a reliable marker of atherosclerosis and peripheral arterial occlusive disease in the present study. Second, renal function was a confounder for the observed associations between serum UA and CV diseases [52]. The findings of the current study may also imply that some mechanisms other than hyperuricemia in CKD may have contributed to a lower ABI and weakened the association between UA and peripheral arterial disease. As for the relationship between UA and arterial stiffness, in a study of 1255 patients with a new diagnosis of hypertension, an independent association between serum UA levels, aortic stiffening and wave reflections was found [53]. In another study of 106 male patients with newly diagnosed type 2 diabetes mellitus, serum UA level was found to be associated with increased aortic and peripheral arterial stiffness [54]. Serum UA may contribute to vascular stiffness through mechanisms such as inducing the production of C-reactive protein (CRP) in vascular endothelial and smooth muscle cells, thereby stimulating cell proliferation [55] and activating inflammatory pathways that promote collagen production and lead to the development of arteriosclerosis [56]. However, we did not find a significant association between UA and baPWV in the present study. In a study of 248 men and 371 women, Cicero et al. found that the association between serum UA and carotid-femoral pulse wave velocity (cfPWV) became non-significant after adjusting for many parameters [57]. The likely reasons for the discrepancies between the present study and previous studies are as follows. First, there were many differences such as the characteristics of the study population, size of the study population, and in the adjustments made for confounding factors between the studies. Some studies included relatively healthy patients or other patient groups without CKD. Second, factors such as hypertension and diabetes mellitus [58] may also have affected arterial stiffness and thus diminished the effect of serum UA in the patients in the present study. However, we further analyzed a subgroup analysis of non-diabetic patients and found the similar results. High UA is still associated with large LAD, large o/p LVM and low LVEF, but is not associated with ABI and baPWV. Therefore, this reason cannot explain the results in this study.

The present study demonstrated the relationships between UA and echocardiographic parameters, ABI and baPWV. It is worth noting that the pathophysiologic pathway of CVD in patients with CKD is complex, and findings of recent studies have helped to clarify the possible mechanism in the pathogenesis of CVD in CKD. In a study of 332 Korean patients with pre-dialysis CKD, Kim et al. found that neutrophil gelatinase-associated lipocalin (NGAL), a biomarker of renal failure and CVD, was associated with LVH and LV diastolic dysfunction. The authors suggested that NGAL is involved in the alterations of the cardiac extracellular matrix, and further contributed to ventricular remodeling [59]. Abnormalities of bone mineral metabolism, such as hyperphosphatemia, hyperparathyroidism, and derangement of vitamin D, are common in patients with CKD even in early stages of the disease, and often complicate matters through dysregulation of bone turnover, mineralization, linear growth and volume. The term CKD-mineral and bone disorders (CKD-MBD) is used to describe such a systemic disorder of mineral and bone metabolism [60]. Previous studies demonstrated that manifestations of CKD-MBD were associated with development of CVD. Clinical and experimental studies have shown that hyperparathyroidism plays a major role in LV dysfunction [61]. Increased fibroblast growth factor-23 (FGF-23) and decreased amounts of its co-receptor, Klotho, were reported to be associated with vascular calcification (VC) in CKD [62]. High phosphate level also contributed to VC by various mechanisms, including transforming vascular smooth muscle cells to an osteochondrogenic phenotype, induction of vascular smooth muscle cells apoptosis, increased FGF-23 level and decreased Klotho expression [63]. Interestingly, recent evidence revealed that UA may be also related to CKD-MBD. UA was found to be positively correlated with parathyroid hormone and FGF-23. In addition, UA can inhibit activity of 1-Alpha-hydroxylase, and further decreased vitamin D levels [64]. These findings should also be taken into consideration when discussing the association between UA and CVD in patients with CKD. However, the data of mineral metabolism is not available in our research. Further study including parameters of mineral metabolism is needed to clarify the relationship between UA and CVD in patients with CKD.

There are several limitations to this study. First, as a cross-sectional study, the results may not precisely reflect the cause–effect relationship between UA and echocardiographic parameters. Second, predicted LVM was calculated using age, sex, height and stroke work. Therefore, single blood pressure measurements may have had large impacts on predicted LVM. An average 24 h ambulatory blood pressure measurement may have been more appropriate to calculate LVM than single clinic blood pressure measurements. Third, the use of antihypertensive medications may have influenced LV geometry and functional parameters. For ethical reasons, we did not withhold any drugs during the study. Therefore, to investigate the influence of the use of medications, we included various classes of antihypertensive drugs in the analysis, and still found that UA was significantly associated with abnormal echocardiographic parameters. In addition, some important data were lacking, such as transmitral flow velocity, mineral metabolism biomarkers, and treatment. Lastly, as previously mentioned, further study including data on mineral metabolism is needed.

In conclusion, in our patients with CKD stages 3–5, we found significant associations among hyperuricemia and high LAD, high o/p LVM and low LVEF. However, a high serum UA level was not closely associated with ABI or baPWV. A high serum UA level may be correlated with abnormal echocardiographic parameters in patients with stage 3–5 CKD. Further longitudinal studies with adjustments for confounding factors are warranted to clarify the relationship between UA and atherosclerosis and arterial stiffness in patients with CKD.

## Figures and Tables

**Figure 1 diagnostics-10-00514-f001:**
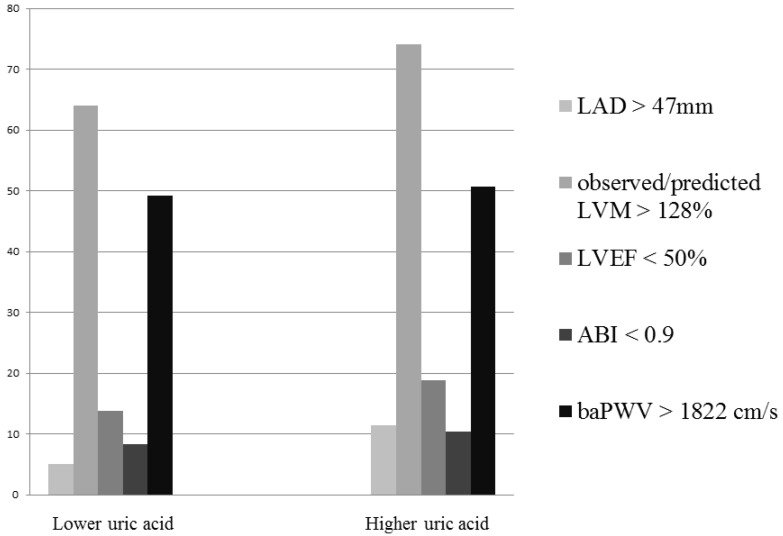
There was a significant higher prevalence of left atrial diameter (LAD) < 47 mm (5.1% vs. 11.4%, *p* = 0.017) and observed/predicted left ventricular mass (LVM) > 128% (64.1% vs. 74.1%, *p* = 0.026) in the high uric acid group, but not left ventricular ejection fraction (LVEF) < 50% (13.8% vs. 18.9%, *p* = 0.160), ankle-brachial index (ABI) < 0.9 (8.3% vs. 10.4%, *p* = 0.450) or brachial-ankle pulse wave velocity (baPWV) > 1822 cm/s (49.3% vs. 50.7%, *p* = 0.769).

**Figure 2 diagnostics-10-00514-f002:**
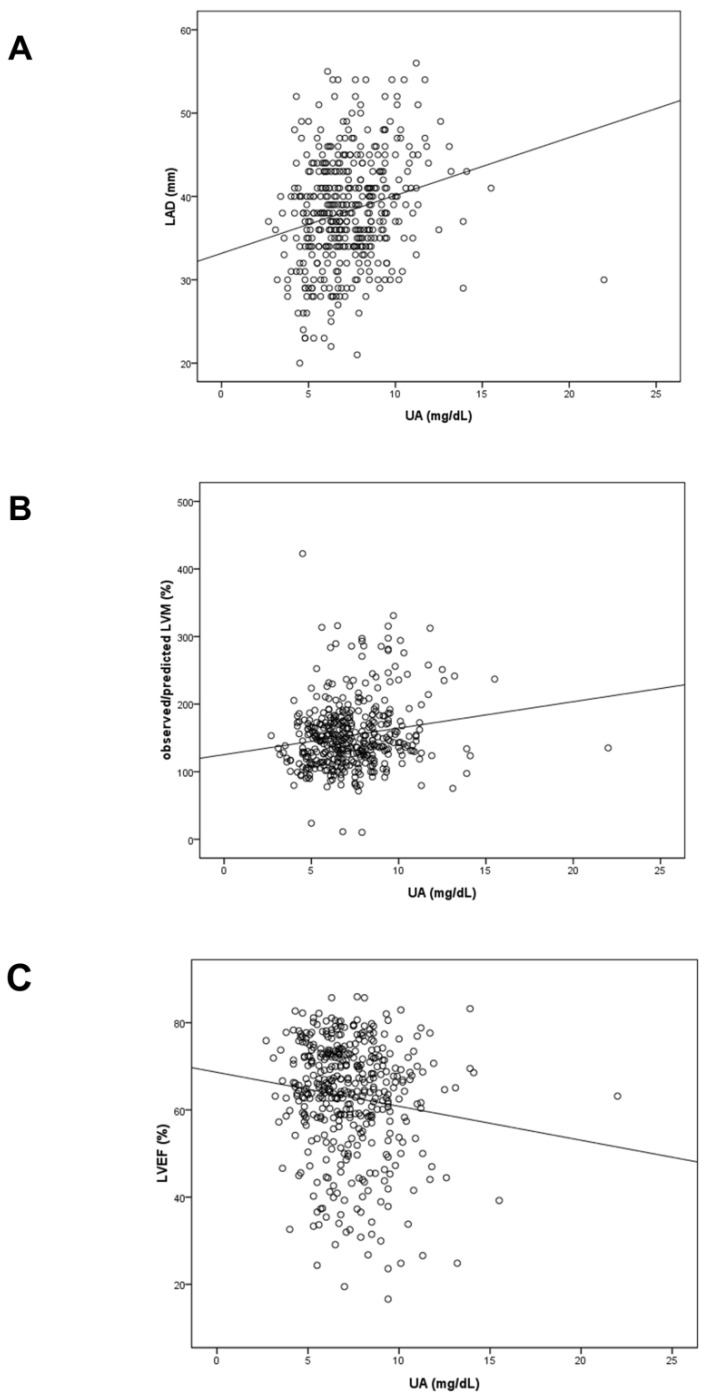
Scatter plots of distribution and correlation of serum UA and LAD (**A**), observed/predicted LVM (**B**) and LVEF (**C**).

**Table 1 diagnostics-10-00514-t001:** Comparison of baseline and echocardiographic characteristics between patients with lower and high uric acid levels.

Characteristics	Lower Uric Acid (*n* = 217)	Higher Uric Acid (*n* = 201)	*p*
Age (year)	65.6 ± 11.5	66.1 ± 13.9	0.665
Male gender (%)	53.9	56.2	0.636
Diabetes mellitus (%)	30.9	38.3	0.110
Coronary artery disease (%)	20.2	19.5	0.861
Gout (%)	8.8	18.9	0.003
Congestive heart failure (%)	4.3	9.0	0.046
Systolic blood pressure (mmHg)	141.2 ± 22.2	139.1 ± 20.8	0.300
Diastolic blood pressure (mmHg)	78.2 ± 12.2	77.4 ± 12.5	0.537
Mean artery pressure (mmHg)	99.2 ± 14.5	98.0 ± 14.2	0.385
ABI	1.10 ± 0.15	1.10 ± 0.15	0.989
baPWV (cm/s)	1903.2 ± 507.9	1869.7 ± 453.2	0.478
Laboratory parameters			
Uric acid (mg/dL)	5.7 ± 1.0	9.0 ± 1.8	< 0.001
Fasting glucose (mg/dL)	114.1 ± 42.0	117.9 ± 48.3	0.429
Triglyceride (mg/dL)	113 (82–169)	128.5 (93.25–199.75)	0.001
Total cholesterol (mg/dL)	193.5 ± 42.1	188.1 ± 41.6	0.203
Hematocrit (%)	39.0 ± 6.8	38.7 ± 7.2	0.707
eGFR (mL/min/1.73 m^2^)	45.7 ± 14.1	38.8 ± 15.7	< 0.001
Medications			
ACEI and/or ARB use (%)	56.2	70.1	0.003
β-blocker use (%)	39.3	49.8	0.033
Calcium channel blocker use (%)	43.3	47.8	0.362
Diuretic use (%)	23.0	43.5	< 0.001
Hypouricemic agent use (%)	7.4	11.4	0.157
Echocardiographic data			
LAD (mm)	37.0 ± 6.5	39.6 ± 6.4	< 0.001
Observed/predicted LVM (%)	146.7 ± 46.1	161.6 ± 54.2	< 0.001
LVEF	64.1 ± 12.4	61.6 ± 14.0	0.055

Abbreviations. UA, uric acid; ABI, ankle-brachial index; baPWV, brachial-ankle pulse wave velocity; eGFR, estimated glomerular filtration rate; ACEI, angiotensin converting enzyme inhibitor; ARB, angiotensin II receptor blocker; LAD, left atrial diameter; LVM, left ventricular mass; LVEF, left ventricular ejection fraction. The study patients were stratified into 2 groups according to sex-specific median values of UA (male: 7.3; female: 6.7 mg/dL).

**Table 2 diagnostics-10-00514-t002:** Determinants of abnormal ABI, baPWV and echocardiographic parameters of study patients using binary logistic regression analysis.

Variables	Multivariable (Forward)	
	OR (95% CI)	*p*
LAD > 47 mm		
Uric acid (per 1 mg/dL)	1.329 (1.111–1.590)	0.002
Observed/predicted LVM > 128%		
Diabetes mellitus	1.934 (1.113–3.361)	0.019
Mean artery pressure (per 1 mmHg)	0.976 (0.958–0.994)	0.009
Uric acid (per 1 mg/dL)	1.198 (1.048–1.369)	0.008
Diuretic use	1.985 (1.099–3.586)	0.023
LVEF < 50%		
Coronary artery disease	2.564 (1.232–5.338)	0.012
Congestive heart failure	6.149 (2.069–18.275)	0.001
Uric acid (per 1 mg/dL)	1.316 (1.111–1.559)	0.002
Fasting glucose (per 1 mg/dL)	1.007 (1.001–1.014)	0.032
Calcium channel blocker use	0.203 (0.092–0.450)	< 0.001
Diuretic use	2.167 (1.045–4.494)	0.038
ABI < 0.9		
Age (per 1 year)	1.160 (1.094–1.231)	0.032
Fasting glucose (per 1 mg/dL)	1.011 (1.004–1.019)	0.003
ACEI and/or ARB use	4.013 (1.123–14.339)	0.032
baPWV > 1822 cm/s		
Age (per 1 year)	1.134 (1.097–1.172)	< 0.001
Diabetes mellitus	3.152 (1.587–6.260)	0.001
Congestive heart failure	0.200 (0.058–0.683)	0.010
Mean artery pressure (per 1 mmHg)	1.132 (1.097–1.168)	< 0.001
Fasting glucose (per 1 mg/dL)	1.013 (1.006–1.020)	< 0.001

Values expressed as odds ratio (OR) and 95% confidence interval (CI). Abbreviations are the same as in Table 1. Covariates in the multivariate model included age, sex, diabetes mellitus, coronary artery disease, congestive heart failure, mean artery pressure, uric acid, fasting glucose, triglyceride, total cholesterol, hematocrit, eGFR and medications use.

**Table 3 diagnostics-10-00514-t003:** Determinants of ABI, baPWV and echocardiographic parameters of study patients using linear regression analysis.

Variables	Multivariable (Stepwise)	
	Unstandardized Coefficient β (95% CI)	*p*
LAD (per 1 mm)		
Congestive heart failure	4.118 (1.466, 6.771)	0.002
Uric acid (per 1 mg/dL)	0.767 (0.421, 1.113)	< 0.001
eGFR (per 1 mL/min/1.73 m^2^)	−0.064 (−0.112, −0.015)	0.010
Observed/predicted LVM (per 1%)		
Age (per 1 year)	−0.435 (−0.800, −0.069)	0.020
Congestive heart failure	63.807 (44.646, 82.969)	< 0.001
Mean artery pressure (per 1 mmHg)	−0.539 (−0.881, −0.197)	0.002
Uric acid (per 1 mg/dL)	4.791 (2.471, 7.111)	< 0.001
Diuretic use	11.522 (1.363, 21.680)	0.026
LVEF (per 1%)		
Coronary artery disease	−4.297 (−7.563, −1.030)	0.010
Congestive heart failure	−12.204 (−17.455, −6.953)	< 0.001
Uric acid (per 1 mg/dL)	−1.126 (−1.763,−0.489)	0.001
Fasting glucose (per 1 mg/dL)	−0.043 (−0.072, −0.015)	0.003
Calcium channel blocker use	5.326 (2.722, 7.929)	< 0.001
ABI (per 0.1)		
Age (per 1 year)	−0.030 (−0.042, −0.019)	< 0.001
Coronary artery disease	−0.389 (−0.745, −0.033)	0.032
Congestive heart failure	−1.169 (−1.745, −0.593)	< 0.001
baPWV (per 10 cm/s)		
Age (per 1 year)	0.184 (0.155, 0.214)	< 0.001
Diabetes mellitus	1.368 (0.478, 2.259)	0.003
Congestive heart failure	−2.750 (−4.292, −1.208)	0.001
Mean artery pressure (per 1 mmHg)	0.173 (0.145, 0.201)	< 0.001
Fasting glucose (per 1 mg/dL)	0.010 (0.001, 0.020)	0.025

Values expressed as unstandardized coefficient βand 95% confidence interval (CI). Abbreviations are the same as in Table 1. Covariates in the multivariate model included age, sex, diabetes mellitus, coronary artery disease, congestive heart failure, mean artery pressure, uric acid, fasting glucose, triglyceride, total cholesterol, hematocrit, eGFR and medications use.

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
