# Peer review of "Hyperuricemia Is Associated with Left Ventricular Dysfunction and Inappropriate Left Ventricular Mass in Chronic Kidney Disease"

_diagnostics, 2020, doi:10.3390/diagnostics10080514_

Round 1

Reviewer 1 Report

  1. In this paper Yu and coworkers investigated in 418 patients with CKD stages 3 to 5  the associations among serum uric acid levels and echocardiographic parameters, ankle-brachial index (ABI), and brachial-ankle pulse wave velocity (baPWV). All these indexes are considered indicators of atherosclerosis and arterial stiffness, in patients with CKD.

In this study, the authors found significant associations among hyperuricemia and high LAD, high observed/predicted LVM and low LVEF. However, ABI and baPWV failed to show significant independent association with serum UA.

Although it is a cross-sectional study (as correctly underlined by the authors themselves) whose results may not precisely reflect the cause-effect relationship between UA and echocardiographic parameters, the paper is quite interesting and provides a valid contribution on this topic also considering that the studies in this regard are few and with contradictory results.  See and cite by the way: Yoshitomi R et al Hypertension Res 2014, II Young Kim et al PLS ONE 2018.

The authors should provide some changes in order to improve the quality and attractiveness of the paper.

- Line 128, the authors state that: Compared to the patients with a lower UA level, those with a higher UA level had a higher prevalence of lower eGFR, used more ACEIs and/or ARBs, β-blockers and diuretics, higher LAD and higher observed/predicted LVM. The authors should comment on this finding and clarify whether this aspect may have influenced the impact of uric acid on echocardiographic parameters.

  • Changes on mineral metabolism begin early in CKD, since patients are in stage 3-5 CKD, the authors should provide and include in their analysis the indexes of mineral metabolisms. If these data are not available, the authors should insert at least some comments in this regard in discussion (see Cozzolino et al Toxins (Basel). 2019 Apr 9; 11 (4): 213. Doi: 10.3390 / toxins11040213 + Cianciolo G et al Curr Vasc Pharmacol. 2018 ) and in case insert this lack among the study limitations.
  • It is presumable that at least stage 4 and 5 patients underwent any therapy in the context of CKD-MBD, therefore which phosphate binder, native Vitamin D, VDRA
  • Finally research efforts are underway to further clarify mechanistic pathways and the role of uric acid in the pathogenesis of CKD-MBD see, cite and comment: Baris Afsar et al Journal of Nephrology https://doi.org/10.1007/s40620-019-  00615-0.

Author Response

In this paper Yu and coworkers investigated in 418 patients with CKD stages 3 to 5  the associations among serum uric acid levels and echocardiographic parameters, ankle-brachial index (ABI), and brachial-ankle pulse wave velocity (baPWV). All these indexes are considered indicators of atherosclerosis and arterial stiffness, in patients with CKD. In this study, the authors found significant associations among hyperuricemia and high LAD, high observed/predicted LVM and low LVEF. However, ABI and baPWV failed to show significant independent association with serum UA.

  1. Although it is a cross-sectional study (as correctly underlined by the authors themselves) whose results may not precisely reflect the cause-effect relationship between UA and echocardiographic parameters, the paper is quite interesting and provides a valid contribution on this topic also considering that the studies in this regard are few and with contradictory results.  See and cite by the way: Yoshitomi R et al Hypertension Res 2014, II Young Kim et al PLS ONE 2018.Ans: Thank you for your comments and providing useful information. We have added new text regarding concept mentioned in two reference articles above. * In a study of 332 Korean patients with pre-dialysis CKD, Kim et al. found that neutrophil gelatinase-associated lipocalin (NGAL), a biomarker of renal failure and CVD, was associated with LVH and LV diastolic dysfunction. The authors suggested that NGAL is involved in the alterations of the cardiac extracellular matrix, and further contributed to ventricular remodeling [59]. (Line 308-312)
  2. * In predialysis patients with CKD stages 3–5, Yoshitomi et al. reported that ABI can predict CV events and mortality and all-cause mortality [46]. (Line 263-265)
  3. The authors should provide some changes in order to improve the quality and attractiveness of the paper.

  1. Line 128, the authors state that: Compared to the patients with a lower UA level, those with a higher UA level had a higher prevalence of lower eGFR, used more ACEIs and/or ARBs, β-blockers and diuretics, higher LAD and higher observed/predicted LVM. The authors should comment on this finding and clarify whether this aspect may have influenced the impact of uric acid on echocardiographic parameters.* In the present study, compared to the patients with a lower UA level, the group with a higher UA level had a higher prevalence of lower eGFR, used more ACEIs and/or ARBs, β-blockers and diuretics. Lower eGFR may contribute to lower UA clearance and higher serum UA level, and the use of antihypertensive medications may have influenced LV geometry and functional parameters. These factors may interfere in the impact of UA on echocardiographic parameters. To investigate the influence of the use of medications and eGFR, we included various classes of antihypertensive drugs and level of eGFR in the multivariate analysis, and still found that UA was significantly associated with abnormal echocardiographic parameters. (Line 239-246)
  2. Ans: Thank you for raising this point. We had added new text discussing the possible impact of medication and eGFR on our findings.

  1. Changes on mineral metabolism begin early in CKD, since patients are in stage 3-5 CKD, the authors should provide and include in their analysis the indexes of mineral metabolisms. If these data are not available, the authors should insert at least some comments in this regard in discussion (see Cozzolino et al Toxins (Basel). 2019 Apr 9; 11 (4): 213. Doi: 10.3390 / toxins11040213 + Cianciolo G et al Curr Vasc Pharmacol. 2018 ) and in case insert this lack among the study limitations.* Abnormalities of bone mineral metabolism, such as hyperphosphatemia, hyperparathyroidism, and derangement of vitamin D, are common in patients with CKD even in early stage of the disease, and often complicate with dysregulation of bone turnover, mineralization, linear growth and volume. CKD-mineral and bone disorders (CKD-MBD) is used to describe such systemic disorder of mineral and bone metabolism [60]. Previous studies had demonstrated that manifestations of CKD-MBD were associated with development of CVD. Clinical and experimental studies had shown that hyperparathyroidism play a major role in LV dysfunction [61]. Increased fibroblast growth factor-23 (FGF-23) and decreased of its co-receptor, Klotho, were reported to be associated with vascular calcification (VC) in CKD [62]. High phosphate level also contributed to VC by various mechanisms, including transforming vascular smooth muscle cells to an osteochondrogenic phenotype, induction of vascular smooth muscle cells apoptosis, increasing FGF-23 level and decreased Klotho expression [63]. (Line 312-323)* Lastly, as previously mentioned, further study including data of mineral metabolism is needed. (Line 342-343)
  2. * Further study including parameters of mineral metabolism is needed to clarify the relationship between UA and CVD in patients with CKD. (Line 328-330)
  3. Ans: Thank you for your kind reminding and providing useful information. The present study is lacking in data of mineral metabolisms. We had added new paragraph about influence of CKD-MBD and added the lack of data into study limitation.
  4. It is presumable that at least stage 4 and 5 patients underwent any therapy in the context of CKD-MBD, therefore which phosphate binder, native Vitamin D, VDRA 
  5. Ans: Thank you for your suggestion. There are 78 patients (18.7%) with stage 4 (n = 41) and 5 (n = 37). There are just few patients with calcium tablets (n = 8), phosphate binder (n = 3), native Vitamin D (n = 0) use, therefore, we did not put into further analysis.
  6. Finally research efforts are underway to further clarify mechanistic pathways and the role of uric acid in the pathogenesis of CKD-MBD see, cite and comment: Baris Afsar et al Journal of Nephrology https://doi.org/10.1007/s40620-019-  00615-0.* Interestingly, recent evidences revealed that UA may be also related to CKD-MBD. UA was found to be positively correlated with parathyroid hormone and FGF-23. In addition, UA can inhibit activity of 1-Alpha-hydroxylase, and further decreased vitamin D levels [64]. These findings should also be taken into consideration when discussing association between UA and CVD in patients with CKD. (Line 323-328)
  7. Ans: Thank you for your kind reminding and providing useful information. We had added new text about the role of uric acid in the pathogenesis of CKD-MBD.

Reviewer 2 Report

Your research is very high quality. It will be interesting to readers.research is very high quality. It will be interesting to readers.

Author Response

Your research is very high quality. It will be interesting to readers.research is very high quality. It will be interesting to readers.

Ans: Thank you for your kindness to read our article.

Reviewer 3 Report

major
In the Discussion part, the authors explain that the lack of significant association between ABI/baPWV and uric acid levels may be due to the high proportion of diabetic patients. Is it possible to support this comment in a subgroup study of non-DM patients?

What was the rate of heart failure among the target patients?  The use of β-blockers and diuretics is high in the high UA group, and the high proportion of patients with heart failure may affect LVM and LAD measurements.

Measurement of the transmitral flow velocity is useful for detecting an increase in left atrial pressure. Did the authors make this measurement? If yes, please provide the data.

Are the authors considering the effects of multicollinearity when adjusting for both SBP and DBP?

The authors adjusted for gout during multivariate analysis. The development of gout usually seems to have a strong correlation with elevated uric acid levels. Is gout a risk factor independent of uric acid levels for the development of cardiovascular disease? Please answer the rational basis for making adjustments with this factor (if you insist that it is necessary to make adjustments with this factor, you do not need to modify the text of manuscript).

Is uric acid level superior in detection sensitivity to iLVM and LAD expansion compared to surface ECG? If you have the data, please show it.

In the conclusions part, the authors state that elevated UA allows early identification of abnormal echocardiographic parameters, but please provide some more data to support this conclusion. For example, please show the distribution and correlation of serum uric acid value and LAD diameter, or iLVM value using a scatter plot. Then consider whether to change the wording of this part to something more modest.

minor
line119 'for triglycerides' Is it typo?

line126 What does AoAC mean?

Author Response

major
1. In the Discussion part, the authors explain that the lack of significant association between ABI/baPWV and uric acid levels may be due to the high proportion of diabetic patients. Is it possible to support this comment in a subgroup study of non-DM patients?

Ans: Thank you for your comments. We further analyzed subgroup analysis of non-DM patients, and found the similar results. High UA is still associated with large LAD, large o/p LVM and low LVEF, whereas not ABI and baPWV. We have added some explanation in the Discussion.

* However, we further analyzed subgroup analysis of non-diabetic patients, and found the similar results. High UA is still associated with large LAD, large o/p LVM and low LVEF, whereas not ABI and baPWV. Therefore, this reason cannot fully explain this result in this study. (Line 302-304)

  1. What was the rate of heart failure among the target patients?  The use of β-blockers and diuretics is high in the high UA group, and the high proportion of patients with heart failure may affect LVM and LAD measurements.

Ans: Thank you for your comments. The prevalence of heart failure was 6.5%. Compared to the patients with a lower UA level, those with a higher UA level had a higher prevalence of heart failure (4.3% vs. 9%, p = 0.046) (Table 1). We have put the variable of heart failure in further analysis in Table 2 and 3.

  1. Measurement of the transmitral flow velocity is useful for detecting an increase in left atrial pressure. Did the authors make this measurement? If yes, please provide the data.

Ans: Thank you for your suggestion. We totally agree that transmitral flow velocity is useful for detecting an increase in left atrial pressure. However, we did not recruit transmitral flow velocity. We put the issue in the Limitation.

* In addition, some echocardiographic data was lacking, such as transmitral flow velocity, which is useful for detecting an increase in left atrial pressure. (Line 340-342)

  1. Are the authors considering the effects of multicollinearity when adjusting for both SBP and DBP?

Ans: Thank you for your comments. To avoid multicollinearity of SBP and DBP, we shift the variables to mean artery pressure.

5.The authors adjusted for gout during multivariate analysis. The development of gout usually seems to have a strong correlation with elevated uric acid levels. Is gout a risk factor independent of uric acid levels for the development of cardiovascular disease? Please answer the rational basis for making adjustments with this factor (if you insist that it is necessary to make adjustments with this factor, you do not need to modify the text of manuscript).

Ans: Thank you for your comments. We totally agreed your point. We did not put the variable of gout into further analysis in Table 2 and 3.

  1. Is uric acid level superior in detection sensitivity to iLVM and LAD expansion compared to surface ECG? If you have the data, please show it. 
  2. Ans: Thank you for your suggestion. However, we did not have the ECG data.
  3. In the conclusions part, the authors state that elevated UA allows early identification of abnormal echocardiographic parameters, but please provide some more data to support this conclusion. For example, please show the distribution and correlation of serum uric acid value and LAD diameter, or iLVM value using a scatter plot. Then consider whether to change the wording of this part to something more modest.
  4. Ans: Thank you for your comments. We have added the figure to show the distribution and correlation of serum uric acid value and LAD, o/pLVM and LVEF using a scatter plot (Figure 2). Besides, we have corrected our sentences in conclusion part and abstract.

* Correlation between UA and echocardiographic parameters

Figure 2 illustrates the scatter plots of distribution and correlation of serum UA and LAD (A), observed/predicted LVM (B) and LVEF (C). The UA was positively correlated with LVEF (r = 0.228, p < 0.001) and observed/predicted LVM (r = 0.167, p = 0.001), and negatively correlated with LVEF (r = -0.127, p = 0.009). (Line 171-178)

* A high serum UA level maybe correlated with abnormal echocardiographic parameters in patients with stage 3-5 CKD. (Line 33-34; Line 346-347)

minor
1. line119 'for triglycerides' Is it typo?

Ans: It is right. We use the value of median (25th-75th percentile) for triglycerides.

  1. line126 What does AoAC mean?

Ans: This sentence is not correct. I have deleted this sentence in the revised manuscript.

Round 2

Reviewer 1 Report

The authors have substantially modified the paper which in my opinion is now eligible for publication. However, I believe that the authors should include the lack of data about mineral metabolism as well as its treatment among the limitations of the study.

Author Response

The authors have substantially modified the paper which in my opinion is now eligible for publication. However, I believe that the authors should include the lack of data about mineral metabolism as well as its treatment among the limitations of the study.

Ans: Thank you for your suggestion. We have added this issue in the Limitation.

* In addition, some important data were lacking, such as transmitral flow velocity, mineral metabolism biomarkers as well as its treatment. (Line 340-342)